# Paternal Diet before Conception and Its Social Determinants in the Elfe Cohort

**DOI:** 10.3390/nu14194008

**Published:** 2022-09-27

**Authors:** Blandine de Lauzon-Guillain, Sacha Krinitzki, Sandrine Lioret, Marie-Aline Charles

**Affiliations:** 1University Paris Cité, Inserm, INRAE, CRESS, 75004 Paris, France; 2Unité Mixte Inserm-Ined-EFS Elfe, Ined, 93300 Aubervilliers, France

**Keywords:** dietary patterns, principal component analysis, fathers, familial characteristics, socioeconomic position

## Abstract

This study aimed to characterize paternal diet during the peri-conception period and its associated characteristics. These cross-sectional analyses were based on 998 fathers from the French nationwide ELFE birth cohort recruited in 2011. Fathers’ diet before mothers’ pregnancies was assessed by a 46-item food frequency questionnaire. Six exploratory dietary patterns were identified by principal component analysis: “Diverse diet”, “Balanced”, “Alcohol”, “Snacking”, “Bread and cheese”, and “Processed products”. Older age was related to higher scores for the “Balanced”, “Alcohol”, and “Snacking” patterns, and high education level with high scores on the “Balanced” pattern and low scores on the “Processed products” pattern. Unemployment and having a first child were related to high scores on the “Alcohol” pattern. Smoking was positively related to “Alcohol” and “Processed products” patterns. A restrictive diet was associated with high scores on the “Balanced” and “Processed products” patterns and low scores on “Alcohol”, “Snacking”, and “Bread and cheese” patterns. Maternal dietary patterns, identified in a previous analysis, were moderately and positively related to the similar patterns among fathers. These findings are important for screening fathers at risk of a suboptimal diet and for accounting for this factor in future studies to examine the specific influence of paternal diet on a child’s health and development.

## 1. Introduction

The developmental origin of health and disease theory postulates that the prenatal environment modulates developmental patterns in the fetus and, hence, affects health and the risk of diseases across the life course [1]. Nutrition is a major determinant of the prenatal environment, and its effect starts even before pregnancy. The maternal pre-conception nutritional status provides nutrients for the early developing embryo and is also able to alter epigenetic processes that are highly active and sensitive early in life [2].

However, the developmental origin of health and disease theory has so far mainly been focused on maternal influences, even though the paternal diet may also induce phenotypic changes in the offspring [3,4]. In animal experiments, paternal food deprivation before conception affected glucose regulation in offspring [5], whereas paternal exposure to an unhealthy diet was related to altered liver function and gut microbiota in offspring [6]. Epigenetics could be an underlying mechanism of this intergenerational transmission [7,8]. Besides nutrition, paternal preconception health and lifestyle have been found to be related to birth defects and malformations in offspring or adverse birth outcomes. For example, paternal smoking before conception and advanced paternal age have been found to be related to low birth weight [9,10,11]. Some studies in humans and animals have examined the relation of paternal alcohol intake before conception to a change in birth weight of the offspring [12,13,14,15,16]. 

However, few human studies have investigated paternal diet during the peri-conception period and its social determinants [17]. One major challenge in the analysis of the association of pre-conception paternal dietary exposure with offspring outcome is the differentiation from maternal dietary factors. Although most studies examined the association between participants’ sex and identified dietary patterns, only a few derived dietary patterns separately for men and women [18,19,20,21,22] and even fewer studies derived dietary patterns separately for fathers and mothers [23,24], especially because father’s dietary data are rarely collected. Among these studies, some dietary patterns were similar between fathers and mothers, but others were specific to one sex [23,24]. Identifying such similarities and discrepancies in diet between mothers and fathers is of great importance for informing future family-based lifestyle and health promotion interventions.

In this context, this study aimed to characterize paternal diet before conception as well as its associations with family characteristics and (di)similarities with maternal diet during pregnancy.

## 2. Materials and Methods

### 2.1. Participants

The present study was based on data from the Etude Longitudinale Française depuis l’Enfance (ELFE) study, a multidisciplinary, nationally representative birth cohort, which included 18,329 children born in a random sample of 349 maternity units in mainland France in 2011. Inclusion took place during 25 selected recruitment days over four waves comprising 4 to 8 days each and covering all four seasons [25]. Inclusion criteria were children born after 33 weeks of gestation to mothers aged 18 years or older and who were not planning to move outside of metropolitan France in the following 3 years. In the fourth wave of the study (December 2011), a specific questionnaire on paternal diet before pregnancy was introduced because of a growing interest in paternal peri-conception diet in the literature. 

### 2.2. Ethics 

Participating mothers had to provide written consent for their own and their child’s participation. When present at inclusion, fathers signed the consent form for the child’s participation or were informed about their rights to oppose it. The ELFE study received approvals from the Advisory Committee for the Treatment of Information on Health Research (Comité Consultatif sur le Traitement des Informations pour la Recherche en Santé), the National Agency Regulating Data Protection (Commission Nationale Informatique et Libertés), and the National Statistics Council.

### 2.3. Family Characteristics

Given that family data were more comprehensively collected during the 2-month interview than during the maternity unit interview and family socio-demographic characteristics only marginally evolved over 2 months, we prioritized data collected at 2 months in our analyses. Socio-demographic characteristics collected during the maternity unit stay were used only in case of missing values at 2 months. 

Parental socio-demographic and socio-economic characteristics studied were paternal and maternal country of birth (France vs. another country), age, education level measured based on the highest academic degree attainment (up to lower secondary school, intermediate, 3-year university degree, and 5-year university degree), and employment status (employed, unemployed, and, for mother, not in the labor force; e.g., housewife, student, disabled), household income per consumption unit, per month (≤€1111, €1112–1500, €1501–1944, €1945–2500, or >€2500), area of residence (rural, urban), region of residence (Ile de France, Mediterranean, North, South, West, East, South East, South West, East Parisian Basin, and West Parisian Basin), and older children in the household (none, 1, or more). To prevent multicolinearity in the present analyses, maternal age and education level were considered relative to paternal values (i.e., maternal age older than the father, same age as the father, 1–2 years younger than the father, 3–4 years younger than the father, at least 5 years younger than the father; maternal educational level higher than the father’s, equivalent, lower level than the father). Paternal smoking was also collected at the 2-month follow-up.

### 2.4. Paternal Diet before Conception

Right after the child’s birth, fathers from the fourth wave (children born in December 2011) were asked to complete an Internet-based self-administered questionnaire, including items on current height and weight, following a restrictive diet in the year before pregnancy, regular physical activity (at least 1 h/week), regular walking (at least 30 min/day, 5 days/week), and a short food frequency questionnaire (FFQ) based on diet in the months before pregnancy (first months of 2011). The consumption frequency of 41 food items, including four non-alcoholic beverages, was scored on a 7-point scale from “never” to “more than once a day”. The consumption frequency of five additional items for alcoholic beverages was assessed by number of glasses consumed per week. Food items were grouped into 30 food groups (Table 1). 

### 2.5. Maternal Diet during Pregnancy

During the maternity unit stay, mothers completed a validated semi-quantitative FFQ on their diet during the last 3 months of their pregnancy; in the present analyses, we used total energy intake (kcal) as well as the scores on five dietary patterns previously identified: “Western diet,” “Balanced diet,” “Bread and toppings,” “Processed products”, and “Milk and breakfast cereals” [26].

Maternal alcohol consumption during pregnancy was not included in the dietary patterns, but in the present paper, we also considered maternal alcohol consumption (none, <1 glass/week, at least 1 glass/week) in early pregnancy (i.e., during the period when she was unknowingly pregnant).

### 2.6. Sample Selection

From the 5590 families included in the fourth wave of the ELFE study, we excluded fathers who did not complete the Internet-based questionnaire (*n* = 4561) and those who completed the questionnaire but with more than 10% (5) missing data (*n* = 19), which resulted in 998 fathers for the identification of dietary patterns and the main analyses. A sensitivity analysis was conducted on a subsample of 926 fathers without any missing data on family- or health-related characteristics.

### 2.7. Statistical Analyses

Comparisons between excluded and included participants involved chi-squared and Student *t* tests.

Dietary patterns were identified by using principal components analysis (PCA), which takes into account the potential effect of interactions within combinations of foods and nutrients. The number of patterns was selected considering eigenvalues > 1, the scree plot, and the interpretability of the patterns. To interpret the results, we considered the items most strongly related to the pattern: those with absolute factor loading value > 0.30. Dietary pattern labels were allocated according to the most significant items associated with the dietary pattern. 

Associations between family characteristics and each paternal dietary pattern were assessed by multivariable linear regressions with a four-step procedure (nested models). The first model included paternal socio-demographic and socio-economic characteristics (i.e., age, education level, employment, country of birth, older children in the household, family income, area of residence). Model 2 further included maternal demographic and socio-economic characteristics (i.e., age, education level, employment, country of birth). Model 3 added paternal health-related characteristics (smoking status, BMI, physical activity, restrictive diet) to model 2. Model 4 further included maternal energy intake, the five maternal dietary patterns, and alcohol consumption in early pregnancy. All models were also adjusted for region of residence and variables related to study design (size of maternity unit).

For fathers with up to four items of missing data on the 46-item questionnaire, missing data on paternal diet were handled with the hot deck method, which replaces the missing value with that for respondents with the same dietary pattern [27]. To deal with missing data for paternal and maternal characteristics, we used multiple imputation with SAS V9.4 (SAS Institute, Cary, NC, USA) to increase power and minimize selection bias in our findings. We assumed that data were missing at random and generated five independent datasets by using the fully conditional specification method (MI procedure, FCS statement, NIMPUTE option). In imputation models, we included all variables of interest after ranking them in ascending order of missing data. Categorical variables were imputed with the multinomial model, ordinal or binary variables with logistic regression, and continuous variables with linear regressions. To generate significance testing of categorical variables, the median of the *p*-values from the imputed data analyses in each dataset was used, as proposed by Eekhout et al. [28]. We also performed a sensitivity analysis on the complete-cases sample. 

All analyses involved using SAS V9.4. *p* < 0.05 was considered statistically significant.

## 3. Results

### 3.1. Sample Characteristics

Among the fathers of a child included in the ELFE study in the fourth wave, as compared with fathers included in the analysis, those excluded from the analysis were more frequently born in a country other than France (abroad: 16.3% vs. 4.8%; *p* < 0.0001), had a lower education level (5-year university degree: 19.0% vs. 28.7%; *p* < 0.0001), had a lower income (€1570 vs. 1837 per consumption unit, per month; *p* < 0.0001), were unemployed (12.3% vs. 9.1%, *p* < 0.0001), smoked (39.5% and 25.4%, *p* < 0.0001), and had older children (56.3% vs. 49.3%; *p* < 0.0001). However, excluded and included fathers did not differ in paternal age (33.1 vs. 33.5 years, *p* = 0.1) nor paternal BMI (26.2 vs. 25.4, *p* = 0.2). The consumption of the different food groups is presented in Appendix A.

### 3.2. Paternal Dietary Patterns before Pregnancy

A total of six dietary patterns were identified for fathers, accounting for 29% of the total variance (Table 1). The first pattern was positively associated with consumption frequency of all food groups and was labelled the “Diverse diet” pattern. The second pattern, labelled “Balanced”, was characterized by high consumption of vegetables, fruit, olive oil, and fish and low consumption of French fries, chips/crackers, processed meat, pizzas and sandwiches, and cakes. The third pattern, labelled “Alcohol”, was defined by high alcohol consumption (wine, beer, and other alcohols) and low consumption of dairy products other than cheese. The fourth pattern, labelled “Snacking”, was characterized by high consumption of cakes, nuts, candies/chocolate, pastries/biscuits, and chips/crackers and low consumption of meat and ham/poultry. The fifth pattern, labelled “Bread and cheese”, was characterized by high consumption of bread/rusks and cheese. The sixth pattern, labelled “Processed products”, was characterized by high consumption of diet products, prepacked products, and other alcohols.

**Table 1 nutrients-14-04008-t001:** Factor loadings for dietary patterns before pregnancy, derived from principal component analysis (*n* = 998): the ELFE study, 2011.

	Factor 1	Factor 2	Factor 3	Factor 4	Factor 5	Factor 6
Ham/Poultry	0.47	−0.11	0.05	−0.34	−0.15	0.27
Meat	0.42	−0.19	0.13	−0.40	0.10	0.03
Candies/Chocolate	0.42	−0.11	−0.28	0.38	0.07	0.03
Prepacked foods	0.42	−0.11	−0.04	−0.01	−0.13	0.37
Processed meat	0.41	−0.35	0.17	−0.14	0.10	−0.03
Potatoes	0.40	−0.08	0.00	−0.23	−0.13	−0.34
Eggs	0.38	0.15	0.02	−0.22	0.03	−0.12
Chips/Crackers	0.38	−0.37	0.18	0.35	−0.10	−0.01
Fish ^1^	0.36	0.33	0.05	−0.19	−0.04	0.04
Pasta/Rice/Semolina	0.31	−0.09	0.00	−0.25	−0.28	−0.18
Cooked vegetables	0.40	0.61	−0.03	0.03	−0.03	−0.14
Fruit	0.33	0.52	−0.16	0.08	−0.15	−0.06
Raw vegetables/salad	0.46	0.51	0.13	0.03	−0.04	−0.06
Olive oil	0.28	0.38	0.14	0.17	0.15	−0.09
Sweetened beverages ^2^	0.22	−0.30	−0.17	0.02	−0.07	−0.01
Pizza/Pies/Sandwiches	0.24	−0.34	0.10	0.04	−0.31	−0.01
French Fries	0.33	−0.40	0.10	−0.17	−0.22	−0.30
Wine	0.05	0.10	0.61	0.22	0.13	0.18
Beer	0.09	−0.01	0.54	0.20	0.06	0.25
Other alcohols ^3^	0.14	−0.18	0.54	0.04	0.08	0.37
Dairy desserts	0.20	−0.17	−0.35	0.04	0.20	0.20
Milk	0.00	−0.05	−0.37	−0.13	0.22	0.23
Dairy products	0.17	0.13	−0.46	−0.16	0.30	0.31
Cakes	0.36	−0.33	−0.25	0.45	0.10	−0.12
Nuts	0.22	0.29	−0.03	0.38	−0.26	−0.09
Pastries/Biscuits	0.33	−0.29	−0.27	0.37	0.19	−0.07
Bread/rusks	0.33	0.03	0.05	−0.16	0.56	−0.12
Cheese	0.29	0.14	0.17	−0.02	0.51	−0.01
Whole grains cereals	0.19	0.22	−0.08	0.20	−0.29	0.25
Diet products ^4^	0.13	0.11	−0.22	−0.11	−0.34	0.54
% variance explained	10%	8%	6%	5%	5%	4%
Label	Diverse diet	Balanced	Alcohol	Snacking	Bread & Cheese	Processed products

^1^ Both canned and fresh; ^2^ including fruit juices; ^3^ short drinks, cider, and strong alcohol; ^4^ diet soda, low sugar products, and low-fat products.

### 3.3. Family Characteristics and Paternal Diet

Bivariable associations between familial characteristics and paternal dietary patterns before conception are presented in Appendix A. 

On multivariable analysis, older fathers were more likely to have high scores on the Balanced, Snacking, Alcohol, and Bread and cheese dietary patterns (Table 2). High paternal education level was related to high scores on the Balanced pattern and low scores on the Processed products pattern. Unemployed fathers scored high on the Alcohol pattern. Fathers born in a country other than France scored low on the Bread and cheese pattern. Having older children in the household was related to low scores on the Alcohol pattern and high scores on the Bread and cheese pattern. Fathers living in low-income households had low scores on the Processed products pattern. Finally, fathers living in rural areas had higher scores on the Diverse diet and Bread and cheese patterns, as compared to those living in urban areas. Maternal demographic and socio-economic characteristics were only marginally related to paternal diet (Table 3). Fathers in a couple relationship with unemployed women and those with lower education than the mother scored high on the Snacking dietary pattern. Age differences between mothers and fathers were related to the father’s Diverse diet pattern, although not linearly.

Concerning health-related characteristics, smoking fathers were more likely to have high scores on the Alcohol pattern (Table 4). Fathers with overweight scored low on the Snacking pattern and high on the Processed products pattern. Regular physical activity was not related to paternal diet, with the exception of a negative association with the Processed products pattern. Conversely, following a restrictive diet in the year before pregnancy was strongly related to almost all dietary patterns, with a positive association with the Balanced or Processed products pattern and a negative association with the Alcohol, Snacking, and Bread and cheese patterns. 

Maternal diet was related to paternal diet (Table 5). Similar patterns for mothers and fathers were the most strongly related, with moderate positive associations between a maternal Western dietary pattern and paternal Diverse diet pattern, between maternal Healthy dietary pattern and paternal Balanced dietary pattern, and between maternal alcohol consumption in early pregnancy and paternal Alcohol pattern. To a lesser extent, the maternal Bread and toppings pattern was positively related to the paternal Bread and cheese pattern, and the maternal Processed products pattern was positively related to the paternal Processed products pattern. The maternal Healthy dietary pattern was also positively related to the paternal Diverse diet pattern, whereas the maternal Western dietary pattern was negatively related to the paternal Balanced dietary pattern.

### 3.4. Sensitivity Analyses

In complete-cases analyses, associations between familial characteristics and paternal dietary patterns remained consistent (data not shown). 

## 4. Discussion

This study provides new insights into fathers’ pre-conception diets, with the following six dietary patterns identified: Diverse diet, Balanced, Alcohol, Snacking, Bread and cheese, and Processed products from a nationwide study (overseas territories excluded). Beyond the well-known positive association of paternal age and education with a balanced diet [23], our findings further suggest that unemployed, older, smoker, and first-time fathers were more likely to adhere to the Alcohol pattern. Younger fathers and those born in a country other than France had low scores on the Bread and cheese pattern, which typically describes the traditional French diet; conversely, non-first-time fathers and rural area dwellers had high scores on this traditional dietary pattern. Additionally, there was some resonance between paternal and maternal dietary patterns, with stronger associations between similar patterns. 

Some studies have investigated dietary patterns across the world but most derived patterns for men and women combined. In France, the INCA2 survey examined dietary patterns of adults (men and women together) and identified five different dietary clusters (“Traditional”, “Prudent”, “Diversified”, and “Processed and sandwiches”) [29]. The INCA2-Prudent diet, characterized by a high consumption of fruits, vegetables, fish, soup, and dietary products, was close to our Balanced pattern. Methodological differences limit the comparison between the INCA2 survey and the present analyses. In the INCA2 survey, dietary clusters were identified for French adults, considering men and women together with age 18–79 years, from a PCA followed by a hierarchical ascending classification, whereas in the ELFE study, we directly used the PCA scores, and our sample was limited to younger men. Few studies have involved young fathers and none in France. Two different dietary studies of fathers of young children, in the United Kingdom (Avon longitudinal study of parents and children [ALSPAC]) [24] and Australia (Melbourne InFANT Program) [23], showed differences in dietary patterns identified as expected due to cultural differences but found patterns similar to the positive and negative parts of our Balanced pattern. 

In studying a potential effect of paternal pre-conception diet on offspring development, associations between paternal and maternal diet need to be disentangled. In the ELFE cohort, information on maternal diet in the last three months of pregnancy was collected by a validated FFQ [26]. Changes in intake in major food groups before and at the end of pregnancy were collected on a four-point scale (increase, no change, decrease, never consumed even before pregnancy), and alcohol intake was estimated at the beginning of pregnancy (before the awareness of being pregnant) and in the last three months of pregnancy. We found the greatest similarities between the peri-conception paternal and pregnancy maternal patterns for patterns based on food eaten during main meals (Diverse diet vs. Western pattern, Healthy vs. Balanced, for fathers and mothers, respectively). These patterns may best capture the shared part of the diet depending on, for example, similar exposure to the foods available at home and shared family meals. The paternal Alcohol pattern was also related to the peri-conception alcohol intake of mothers. Considering both intakes of alcohol on fetal development in further analyses would be of interest. Snacking may result from a more personal dietary behavior, but there is some association between the fathers’ Snacking and Bread and cheese patterns and the mothers’ Bread and toppings pattern, which reflects the contribution of the bread food group to snacking patterns in France. The weak association between the maternal Bread and toppings pattern and the paternal Bread and cheese pattern is probably due to the construction of the patterns in that it was mainly not only related to bread and cheese consumption for fathers but also included sweet products that are often consumed with bread in France, such as chocolate and honey/jam for mothers. Likewise, the Processed products pattern was quite different among fathers and mothers in part because ready-prepared dishes, a major component of this pattern for mothers, was not assessed in fathers. Finally, the paternal Balanced pattern was related positively to the maternal Healthy pattern and negatively to the maternal Western pattern in that it was, by construction, related to high consumption of healthy foods and low consumption of foods typical of the Western diet. Similarly, in both the ALSPAC and the InFANT studies, the highest correlations between men and women were for the healthy/unhealthy dietary patterns [23,24].

The Balanced pattern was related to family characteristics consistent with previous studies: fathers with high scores on this pattern were older, had a higher education level, and tended to be more physically active in their spare time than those with lower score [22,24,29,30]. Additionally, men who reported drinking alcohol before the pregnancy were more likely to be older and smokers [15]. However, the present study also highlighted that they were more likely to be unemployed and first-time fathers. Migration status was only related to the more traditional dietary pattern, fathers born in a country other than France being less likely to have high scores on this pattern. This association is consistent with previous findings for pregnant women [26] and suggests that this pattern is the most sensitive to the cultural differences. Interestingly, younger fathers were also less likely to have high scores on this pattern, suggesting that more traditional dietary pattern are less adopted by younger fathers. These characteristics need to be taken into account in planning interventions focusing on paternal diet.

Although the ELFE study is a large nationwide birth cohort, this analysis is based on a questionnaire introduced only in the last wave of recruitment, and there was no attempt to remind non-responding fathers. Only about one fifth of eligible fathers answered the questionnaire. As this is the first time this type of study has been carried out in France, the study did not seek to be representative but rather aimed to obtain information on the diet of the parental dyad in a sample with sufficient variability to be able to study possible specificities of the paternal diet. Furthermore, the information on the paternal diet was collected by using a short not-validated FFQ, that did not allow for the calculation of nutrient intake, and we have to acknowledge a potential memory bias because fathers were asked about their diet about one year before completing the questionnaire. However, the objective of this study was not to estimate energy or nutrient intake but to rank fathers according to their patterns of food intake, which was proven efficient according to the FFQ. The variability of the reported paternal diet was sufficient to identify six different patterns, and the findings reported in this paper were consistent with those previously collected on maternal diet during pregnancy by use of a validated FFQ [26]. Missing data on dietary items and sociodemographic variables were addressed by the multiple imputation method [31], and sensitivity analyses based on completed cases showed very consistent results. 

## 5. Conclusions

To conclude, besides the association between paternal age or education and diet quality, some characteristics were strongly related to paternal diet: the area of residence and the country of birth, unemployment, and the presence of older children in the household. These must be accounted for when planning nutritional interventions. The moderate correlations of similar dietary patterns between fathers and mothers could enhance the benefits of such programs at the family level, although the specificities of their respective diets are worth accounting for. Given the known impact of nutrition on epigenetics, examining the potential influence of paternal diet in the peri-conception period on child’s post-natal development and metabolism will be of major interest. Paternal and maternal dietary patterns have enough dissimilarities to be studied simultaneously with offspring development. 

## Figures and Tables

**Table 2 nutrients-14-04008-t002:** Multivariable associations between paternal and household characteristics and paternal diet (*n* = 998): the ELFE study, 2011.

	Diverse Diet	Balanced	Alcohol	Snacking	Bread and Cheese	Processed Products
Paternal characteristics						
Age, years						
<27	0.14 [−0.12; 0.41] ^†^	−0.18 [−0.42; 0.07]	−0.31 [−0.57; −0.04]	−0.16 [−0.42; 0.10]	−0.25 [−0.51; 0.01]	0.02 [−0.24; 0.28]
27–31	0 [Ref]	0 [Ref]	0 [Ref]	0 [Ref]	0 [Ref]	0 [Ref]
32–36	0.03 [−0.13; 0.19]	0.27 [0.12; 0.42]	−0.02 [−0.18; 0.14]	0.09 [−0.07; 0.25]	−0.02 [−0.18; 0.14]	−0.01 [−0.17; 0.16]
≥37	−0.05 [−0.23; 0.13]	0.57 [0.41; 0.74]	0.13 [−0.04; 0.31]	0.18 [0.00; 0.36]	0.01 [−0.17; 0.19]	−0.01 [−0.19; 0.17]
Education level						
Up to upper secondary school	−0.09 [−0.28; 0.10]	−0.45 [−0.63; −0.27]	0.00 [−0.18; 0.18]	−0.12 [−0.30; 0.07]	0.13 [−0.05; 0.31]	0.23 [0.05; 0.41]
Intermediate	0.00 [−0.20; 0.19]	−0.30 [−0.48; −0.11]	−0.03 [−0.22; 0.17]	−0.12 [−0.31; 0.08]	0.11 [−0.09; 0.30]	0.23 [0.03; 0.42]
3-year university degree	0.11 [−0.10; 0.33]	−0.07 [−0.26; 0.13]	−0.02 [−0.24; 0.19]	0.02 [−0.18; 0.23]	0.00 [−0.21; 0.21]	0.07 [−0.14; 0.28]
At least 5-year university degree	0 [Ref]	0 [Ref]	0 [Ref]	0 [Ref]	0 [Ref]	0 [Ref]
Country of birth						
Born abroad	−0.17 [−0.46; 0.13]	0.12 [−0.16; 0.40]	0.00 [−0.29; 0.30]	0.12 [−0.18; 0.42]	−0.42 [−0.71; −0.13]	−0.14 [−0.43; 0.16]
Born in France	0 [Ref]	0 [Ref]	0 [Ref]	0 [Ref]	0 [Ref]	0 [Ref]
Older children in household						
First child	0 [Ref]	0 [Ref]	0 [Ref]	0 [Ref]	0 [Ref]	0 [Ref]
At least one other child	0.12 [−0.02; 0.26]	−0.01 [−0.14; 0.12]	−0.20 [−0.34; −0.06]	−0.11 [−0.25; 0.03]	0.15 [0.02; 0.29]	−0.07 [−0.21; 0.06]
Employment						
Employed	0 [Ref]	0 [Ref]	0 [Ref]	0 [Ref]	0 [Ref]	0 [Ref]
Unemployed	0.18 [−0.04; 0.41]	−0.02 [−0.23; 0.19]	0.24 [0.02; 0.46]	0.00 [−0.23; 0.22]	−0.04 [−0.26; 0.18]	0.00 [−0.22; 0.22]
Household characteristics						
Income per consumption unit, per month						
<€1112	0.04 [−0.20; 0.28]	−0.18 [−0.40; 0.04]	0.12 [−0.12; 0.36]	−0.05 [−0.28; 0.18]	−0.19 [−0.41; 0.03]	−0.28 [−0.53; −0.03]
€1112–1500	0 [Ref]	0 [Ref]	0 [Ref]	0 [Ref]	0 [Ref]	0 [Ref]
€1501–1944	0.06 [−0.12; 0.23]	0.00 [−0.17; 0.17]	−0.07 [−0.24; 0.11]	−0.09 [−0.27; 0.09]	−0.08 [−0.26; 0.09]	−0.01 [−0.19; 0.17]
€1945–2500	−0.12 [−0.33; 0.08]	0.08 [−0.12; 0.27]	−0.13 [−0.33; 0.07]	−0.07 [−0.27; 0.13]	−0.09 [−0.29; 0.10]	0.04 [−0.16; 0.24]
>€2500	−0.16 [−0.40; 0.09]	0.13 [−0.11; 0.37]	−0.14 [−0.38; 0.10]	−0.11 [−0.36; 0.14]	0.02 [−0.22; 0.27]	0.11 [−0.13; 0.36]
City size						
Rural area	0.20 [0.05; 0.35]	−0.11 [−0.25; 0.04]	−0.07 [−0.23; 0.08]	0.05 [−0.10; 0.21]	0.26 [0.11; 0.41]	0.06 [−0.09; 0.21]
Urban area	0 [Ref]	0 [Ref]	0 [Ref]	0 [Ref]	0 [Ref]	0 [Ref]

^†^ Values are estimates [95% CI] from linear regressions considering simultaneously all paternal and household characteristics and also adjusted for region of residence and maternity unit size.

**Table 3 nutrients-14-04008-t003:** Multivariable associations between maternal characteristics and paternal diet (*n* = 998): the ELFE study, 2011.

	Diverse Diet	Balanced	Alcohol	Snacking	Bread and Cheese	Processed Products
Maternal characteristics						
Age difference with father						
Older mother	0.27 [0.06; 0.48] ^†^	0.05 [−0.15; 0.25]	−0.06 [−0.27; 0.15]	0.03 [−0.18; 0.24]	−0.05 [−0.25; 0.16]	−0.08 [−0.29; 0.13]
Same age	0 [Ref]	0 [Ref]	0 [Ref]	0 [Ref]	0 [Ref]	0 [Ref]
Mother 1–2 years younger	0.19 [−0.03; 0.40]	0.08 [−0.12; 0.28]	−0.13 [−0.34; 0.09]	0.02 [−0.19; 0.23]	0.00 [−0.20; 0.21]	−0.07 [−0.28; 0.14]
Mother 3–4 years younger	0.27 [0.04; 0.51]	−0.09 [−0.31; 0.13]	−0.09 [−0.32; 0.14]	0.18 [−0.05; 0.41]	−0.02 [−0.24; 0.21]	−0.24 [−0.47; −0.01]
Mother at least 5 years younger	0.09 [−0.19; 0.37]	−0.17 [−0.43; 0.08]	−0.09 [−0.36; 0.18]	0.06 [−0.22; 0.33]	−0.17 [−0.44; 0.11]	0.00 [−0.27; 0.26]
Education level difference with father						
Equivalent level	0 [Ref]	0 [Ref]	0 [Ref]	0 [Ref]	0 [Ref]	0 [Ref]
Father’s > Mother’s level	−0.15 [−0.34; 0.03]	−0.08 [−0.26; 0.09]	−0.03 [−0.21; 0.15]	−0.02 [−0.20; 0.16]	0.10 [−0.10; 0.29]	−0.01 [−0.19; 0.17]
Father’s < Mother’s level	0.06 [−0.11; 0.22]	0.06 [−0.09; 0.21]	−0.03 [−0.20; 0.13]	0.23 [0.07; 0.40]	0.08 [−0.08; 0.24]	−0.14 [−0.31; 0.03]
Country of birth						
Born abroad	−0.08 [−0.37; 0.21]	0.08 [−0.19; 0.35]	0.06 [−0.23; 0.35]	−0.16 [−0.45; 0.13]	−0.13 [−0.41; 0.15]	−0.06 [−0.35; 0.23]
Born in France	0 [Ref]	0 [Ref]	0 [Ref]	0 [Ref]	0 [Ref]	0 [Ref]
Employment during pregnancy						
Employed	0 [Ref]	0 [Ref]	0 [Ref]	0 [Ref]	0 [Ref]	0 [Ref]
Unemployed	0.00 [−0.22; 0.22]	0.11 [−0.09; 0.31]	0.07 [−0.15; 0.29]	0.29 [0.07; 0.51]	0.02 [−0.20; 0.23]	0.07 [−0.15; 0.28]
Out of the labor force	−0.12 [−0.34; 0.11]	0.07 [−0.14; 0.28]	−0.15 [−0.37; 0.08]	0.01 [−0.22; 0.24]	0.01 [−0.22; 0.23]	−0.06 [−0.30; 0.17]

^†^ Values are estimates [95% CI] from linear regressions considering simultaneously all maternal characteristics and adjusted for paternal and household characteristics, region of residence, and maternity unit size.

**Table 4 nutrients-14-04008-t004:** Multivariable associations between paternal health-related characteristics and paternal diet (*n* = 998): the ELFE study, 2011.

	Diverse Diet	Balanced	Alcohol	Snacking	Bread and Cheese	Processed Products
Health-related characteristics						
Smoking status						
No smoker	0 [Ref]	0 [Ref]	0 [Ref]	0 [Ref]	0 [Ref]	0 [Ref]
Smoker	−0.07 [−0.23; 0.10] ^†^	−0.08 [−0.23; 0.08]	0.50 [0.35; 0.66]	0.00 [−0.15; 0.15]	0.01 [−0.15; 0.16]	0.19 [0.04; 0.35]
BMI						
<18.5 kg/m^2^	0.10 [−0.65; 0.85]	−0.15 [−0.85; 0.56]	−0.26 [−0.99; 0.47]	0.54 [−0.20; 1.28]	0.37 [−0.37; 1.10]	−0.26 [−0.98; 0.47]
18.5–24.9 kg/m^2^	0 [Ref]	0 [Ref]	0 [Ref]	0 [Ref]	0 [Ref]	0 [Ref]
25–29.9 kg/m^2^	−0.05 [−0.19; 0.09]	−0.02 [−0.15; 0.11]	−0.02 [−0.16; 0.12]	−0.19 [−0.33; −0.05]	−0.01 [−0.15; 0.13]	0.12 [−0.02; 0.25]
At least 30 kg/m^2^	−0.10 [−0.33; 0.13]	−0.10 [−0.32; 0.12]	0.12 [−0.11; 0.35]	−0.11 [−0.34; 0.13]	−0.20 [−0.42; 0.03]	0.18 [−0.04; 0.41]
Regular physical activity ^1^						
No	0 [Ref]	0 [Ref]	0 [Ref]	0 [Ref]	0 [Ref]	0 [Ref]
Yes	0.12 [−0.02; 0.25]	0.11 [−0.02; 0.23]	0.06 [−0.07; 0.20]	0.06 [−0.08; 0.20]	0.04 [−0.09; 0.17]	−0.19 [−0.32; −0.06]
Regular walking ^2^						
No	0 [Ref]	0 [Ref]	0 [Ref]	0 [Ref]	0 [Ref]	0 [Ref]
Yes	0.04 [−0.09; 0.17]	0.12 [−0.01; 0.24]	0.02 [−0.11; 0.15]	0.00 [−0.13; 0.14]	−0.02 [−0.15; 0.11]	0.04 [−0.09; 0.16]
Restrictive diet						
No	0 [Ref]	0 [Ref]	0 [Ref]	0 [Ref]	0 [Ref]	0 [Ref]
Yes	0 [−0.28; 0.28]	0.51 [0.25; 0.77]	−0.29 [−0.57; −0.02]	−0.50 [−0.78; −0.23]	−0.41 [−0.69; −0.14]	0.93 [0.66; 1.20]

^†^ Values are estimates [95% CI] from linear regressions considering simultaneously all maternal characteristics and adjusted for paternal and household characteristics, maternal characteristics, region of residence, and maternity unit size; ^1^ at least 1 h/week; ^2^ at least 30 min/day, 5 days/week.

**Table 5 nutrients-14-04008-t005:** Multivariable associations between maternal diet and paternal diet (*n* = 998): the ELFE study, 2011.

	Diverse Diet	Balanced	Alcohol	Snacking	Bread and Cheese	Processed Products
Maternal diet during pregnancy						
Energy intake (per 100 kcal)	−0.01 [−0.03; 0.01] ^†^	−0.01 [−0.03; 0.00]	−0.01 [−0.02; 0.01]	0.00 [−0.02; 0.02]	0.01 [−0.01; 0.03]	0.00 [−0.02; 0.02]
PCA pattern 1: Western diet	0.32 [0.17; 0.48]	−0.17 [−0.30; −0.04]	0.06 [−0.08; 0.20]	−0.07 [−0.22; 0.09]	−0.24 [−0.39; −0.10]	−0.09 [−0.24; 0.05]
PCA pattern 2: Healthy diet	0.15 [0.07; 0.24]	0.36 [0.28; 0.43]	−0.02 [−0.10; 0.06]	−0.02 [−0.11; 0.06]	−0.05 [−0.12; 0.03]	−0.01 [−0.09; 0.07]
PCA pattern 3: Bread and toppings	0.03 [−0.06; 0.12]	0.14 [0.06; 0.21]	0.00 [−0.10; 0.09]	0.12 [0.02; 0.21]	0.17 [0.08; 0.25]	0.09 [0.00; 0.17]
PCA pattern 4: Processed products	−0.01 [−0.09; 0.07]	−0.07 [−0.15; 0.01]	0.03 [−0.05; 0.11]	0.05 [−0.02; 0.13]	−0.04 [−0.11; 0.04]	0.15 [0.07; 0.22]
PCA pattern 5: Milk and breakfast cereals	−0.04 [−0.13; 0.04]	0.05 [−0.02; 0.12]	−0.11 [−0.18; −0.03]	−0.03 [−0.11; 0.04]	−0.03 [−0.10; 0.04]	0.17 [0.11; 0.24]
Alcohol in early pregnancy						
Never during this period	0 [Ref]	0 [Ref]	0 [Ref]	0 [Ref]	0 [Ref]	0 [Ref]
<1 glass/week	0.13 [−0.02; 0.29]	0.07 [−0.07; 0.21]	0.16 [−0.01; 0.32]	0.03 [−0.13; 0.19]	−0.12 [−0.27; 0.04]	0.13 [−0.03; 0.29]
At least 1 glass/week	0.17 [0.00; 0.33]	0.15 [0.00; 0.30]	0.45 [0.28; 0.61]	0.12 [−0.05; 0.29]	0.08 [−0.08; 0.24]	0.16 [0.00; 0.32]

^†^ Values are estimates [95% CI] from linear regressions considering simultaneously all maternal dietary patterns and adjusted for paternal and household characteristics, maternal characteristics, paternal health-related characteristics, region of residence, and maternity unit size.

## Data Availability

The data underlying the findings cannot be made freely available for ethical and legal restrictions imposed, because this study includes a substantial number of variables that, together, could be used to re-identify the participants based on a few key characteristics and then be used to have access to other personal data. Therefore, the French ethics authority strictly forbids making these data freely available. However, they can be obtained upon request from the ELFE principal investigator. Readers may contact marie-aline.charles@inserm.fr to request the data.

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
