# Peer review of "Paternal Diet before Conception and Its Social Determinants in the Elfe Cohort"

_nutrients, 2022, doi:10.3390/nu14194008_

Round 1

Reviewer 1 Report

The manuscript “Paternal diet before conception and its social determinants in the ELFE cohort” estimated crude qualitative parameters of dietary patterns and attempted to paint its implication in the finer outcome of offspring development. Although the manuscript promises to deliver a piece of new knowledge, the take-home message failed to convey to the readers. 

While authors hypothesize that paternal diet could contribute to the associated factors of maternal diet and its interaction with fetal growth and development, the study title and objectives are not mentioned. The outcome measurement of the paternal diet and its social determinant is missing in the study title. Similarly, the aim of the study is not written in the abstract.

The study used “energy” as one of the components in PCA but did not calculate the calorie values of foods from FFQ. It is possible to calculate crude values of dietary components into energy. The author should consider the differential energy intake in the parental and maternal diet.

Refer to Supplementary table 1. Descriptive statistics on the paternal diet - Jam/Poultry are foods primarily enriched with different micronutrients and energy; How could these be used in a similar category in food frequency?

The contribution of the paternal diet before conception was established for genome imprinting for several attributes of growth and phenotypes of the offspring. Thus, in addition to food, lifestyle factors like exercise, smoking and alcohol intake could have independent effects. If alcohol is a principal component in dietary analysis, its energy contribution should be counted. The discussion should have included such limitations.

Author Response

The manuscript “Paternal diet before conception and its social determinants in the ELFE cohort” estimated crude qualitative parameters of dietary patterns and attempted to paint its implication in the finer outcome of offspring development. Although the manuscript promises to deliver a piece of new knowledge, the take-home message failed to convey to the readers. 

While authors hypothesize that paternal diet could contribute to the associated factors of maternal diet and its interaction with fetal growth and development, the study title and objectives are not mentioned. The outcome measurement of the paternal diet and its social determinant is missing in the study title. Similarly, the aim of the study is not written in the abstract.

Response: The objective of the present study was first to describe paternal diet and then to identify family characteristics (including maternal own diet) that were related to paternal diet, as indicated in the abstract “This study aimed to characterize paternal diet during the peri-conception period and its associated characteristics.” and at the end of the introduction section “In this context, this study aimed to characterize paternal diet before conception as well as its associations with family characteristics and (di)similarities with maternal diet during pregnancy. Both the exposure and the outcome are therefore indicated in the title.

This is a first step before examining the independent effect of paternal diet on child health and development. This specific analysis was however beyond the scope of the current paper, thus mentioned as a research perspective in the conclusion of the paper “Given the known impact of nutrition on epigenetics, examining the potential influence of paternal diet in the peri-conception period on child's post-natal development and metabolism will be of major interest. Paternal and maternal dietary patterns have enough dissimilarities to be studied simultaneously with offspring development.”

The study used “energy” as one of the components in PCA but did not calculate the calorie values of foods from FFQ. It is possible to calculate crude values of dietary components into energy. The author should consider the differential energy intake in the parental and maternal diet.

Réponse: Unfortunately, the food frequency questionnaire used in the present study was semi-quantitative thus did not allow to assess energy intake. We acknowledged more clearly this limitation in the revised manuscript “Furthermore, the information on the paternal diet was collected by using a short not-validated FFQ, that did not allow for the calculation of nutrient intake, and we have to acknowledge a potential memory bias because fathers were asked about their diet about 1 year before completing the questionnaire. However, the objective of this study was not to estimate energy or nutrient intake but to rank fathers according to their patterns of food intake, which was proven efficient according to the FFQ.”

The first dietary pattern positively correlated to all food items. Then, we considered that father with higher scores on this pattern were those with the higher energy intake. That’s why we first decided to label it “Energy”. As it could be misleading, we modified, in the revised version of the paper, the label as follows “Diverse diet”.

Refer to Supplementary table 1. Descriptive statistics on the paternal diet - Jam/Poultry are foods primarily enriched with different micronutrients and energy; How could these be used in a similar category in food frequency?

Response: We apologize for the typing error. It has been corrected to ham/poultry.

The contribution of the paternal diet before conception was established for genome imprinting for several attributes of growth and phenotypes of the offspring. Thus, in addition to food, lifestyle factors like exercise, smoking and alcohol intake could have independent effects. If alcohol is a principal component in dietary analysis, its energy contribution should be counted. The discussion should have included such limitations.

Response: We agree with the reviewer that other paternal lifestyle factors have been associated with epigenetic changes in the sperms and also with offspring health. This was stated in the introduction “Besides nutrition, paternal preconception health and lifestyle have been found related to birth defects and malformations in offspring or adverse birth outcomes”. In the present paper, we did not relate paternal diet to child’s outcomes, though it is a perspective of this work. Of course, it will be important to consider other father’s behaviors related to energy balance and father’s lifestyle habits when we will examine to which extent the father's diet influences child’s health and development.

Reviewer 2 Report

This study examined the associations between paternal dietary patterns (e.g., energy, balanced, alcohol, etc.) and paternal/household characteristics (e.g., area of residence, country of birth, employment and having other children). The authors also performed correlations with maternal dietary patterns and found enough dissimilarities to suggest further investigating paternal dietary patterns on offspring development. There is a lack of study in this field in humans and epigenetics findings in animals suggest that paternal diet should be further investigated to deepen our understanding of the impact of paternal diet on offspring’s future health. This could potentially help to better adapted nutritional interventions and guide decision-makers in influencing the determinants of paternal dietary practices (policies, social norms, food marketing, etc.).  

Although there are important drawbacks to the study such as using a non-validated short food frequency questionnaire, having a non-representative sample of fathers and the time difference between the diet recall (before birth) and data collection (after birth), those issues are appropriately highlighted in the discussion section.

Minor comments:

p.3 line 99 – I would suggest removing the word “retired” since it is unlikely to be the case in your cohort and is not shown in your table 3.

Tables 2, 3 and 4. the asterisks are not aligned with numbers, so it is difficult to see what is significant. Please arrange like in table 5.

P.11 – “Qualitatively similar patterns for 251 mothers and … ». This qualitative analysis is also mentioned in the conclusion (lines 360-361).

- What do you mean by “qualitatively”? This does not refer to qualitative methods (e.g., interviews, focus group) so how was it established that the patterns were qualitatively similar? Please explain in the methods section.

P. 15. “Migration status was poorly related to fathers’ dietary pattern. »

- It would be interesting to comment this sentence, otherwise as it stands, it leaves us on our appetite.

P. 15” To conclude, this is the first time this type of study has been carried out in France.”

- This could go in a section on the “strengths” of the study rather than being the first sentence of the conclusion. The following sentences better reflect a summary/conclusion of the study.

Author Response

This study examined the associations between paternal dietary patterns (e.g., energy, balanced, alcohol, etc.) and paternal/household characteristics (e.g., area of residence, country of birth, employment and having other children). The authors also performed correlations with maternal dietary patterns and found enough dissimilarities to suggest further investigating paternal dietary patterns on offspring development. There is a lack of study in this field in humans and epigenetics findings in animals suggest that paternal diet should be further investigated to deepen our understanding of the impact of paternal diet on offspring’s future health. This could potentially help to better adapted nutritional interventions and guide decision-makers in influencing the determinants of paternal dietary practices (policies, social norms, food marketing, etc.).  

Although there are important drawbacks to the study such as using a non-validated short food frequency questionnaire, having a non-representative sample of fathers and the time difference between the diet recall (before birth) and data collection (after birth), those issues are appropriately highlighted in the discussion section.

Minor comments: 

p.3 line 99 – I would suggest removing the word “retired” since it is unlikely to be the case in your cohort and is not shown in your table 3. 

Response: As suggested, the word “retired” has been removed.

Tables 2, 3 and 4. the asterisks are not aligned with numbers, so it is difficult to see what is significant. Please arrange like in table 5. 

Response: The asterisks were not aligned with numbers as they represented the global (type III) p-value of the categorical variables and not the significance of each category. As it could be misleading, we decided to remove these columns, the significance of each category being already given with the 95% confidence intervals.

P.11 – “Qualitatively similar patterns for 251 mothers and … ». This qualitative analysis is also mentioned in the conclusion (lines 360-361).

- What do you mean by “qualitatively”? This does not refer to qualitative methods (e.g., interviews, focus group) so how was it established that the patterns were qualitatively similar? Please explain in the methods section. 

Response: The use of the word “qualitative” was misused to indicate that the answer to the question was categorical and not quantitative. We did not used qualitative methods in our study. To avoid misinterpretation, we decided to remove this word from the conclusion and the sentence in the discussion has been modified as follows: “Changes in intake in major food groups before and at the end of pregnancy were collected on a 4-point scale (increase, no change, decrease, never consumed even before pregnancy)”.

  1. 15. “Migration status was poorly related to fathers’ dietary pattern. » 

- It would be interesting to comment this sentence, otherwise as it stands, it leaves us on our appetite. 

Response: This part of the discussion has been modified as follows: “This association is consistent with previous findings for pregnant women [26], and suggests that this pattern is the most sensitive to the cultural differences. Interestingly, younger fathers were also less likely to have high scores on this pattern, suggesting that more traditional dietary pattern are less (or not yet) adopted by younger fathers.

  1. 15” To conclude, this is the first time this type of study has been carried out in France.” 

- This could go in a section on the “strengths” of the study rather than being the first sentence of the conclusion. The following sentences better reflect a summary/conclusion of the study. 

Response: The sentence has been moved to the last paragraph of the discussion.